# Understanding and Exploring the Network with Stochastic Architectures

**Zhijie Deng, Yinpeng Dong, Shifeng Zhang, Jun Zhu***

Dept. of Comp. Sci. & Tech., Institute for AI, BNRist Center
Tsinghua-Bosch Joint ML Center, THBI Lab, Tsinghua University, Beijing, 100084 China
`{dzj17,dyp17,zhangsf15}@mails.tsinghua.edu.cn, dcszj@mail.tsinghua.edu.cn`

## Abstract

There is an emerging trend to train a network with stochastic architectures to enable various architectures to be plugged and played during inference. However, the existing investigation is highly entangled with neural architecture search (NAS), limiting its widespread use across scenarios. In this work, we decouple the training of a network with stochastic architectures (NSA) from NAS and provide a first systematical investigation on it as a stand-alone problem. We first uncover the characteristics of NSA in various aspects ranging from training stability, convergence, predictive behaviour, to generalization capacity to unseen architectures. We identify various issues of the vanilla NSA, such as training/test disparity and function mode collapse, and further propose the solutions to these issues with theoretical and empirical insights. We believe that these results could also serve as good heuristics for NAS. Given these understandings, we further apply the NSA with our improvements into diverse scenarios to fully exploit its promise of inference-time architecture stochasticity, including model ensemble, uncertainty estimation and semi-supervised learning. Remarkable performance (*e.g.*, 2.75% error rate and 0.0032 expected calibration error on CIFAR-10) validate the effectiveness of such a model, providing new perspectives of exploring the potential of the network with stochastic architectures, beyond NAS.

## 1 Introduction

Deep neural networks (DNNs) are the *de facto* methods to model complex data in a wide spectrum of practical scenarios [12, 36, 38, 40]. The design of neural architectures has always been an active research topic in DNNs, aiming to discover effective connectivity patterns for building networks, in manually designed [33, 14, 43, 13, 51, 32] or automatic [52, 31, 53, 22] manners. Recent research even permits us to train a network without a fixed architecture [3, 45, 42, 1, 11], *i.e.*, at every training iteration, an architecture sample is randomly drawn from an architecture distribution and used to guide the training of network weights (see Fig. 1 for more insights), which is also known as the weight sharing technique in neural architecture search (NAS)[1].

Though the weight-sharing network with *stochastic architectures* is promising, its usage is closely encoupled with NAS to relieve the burden of training thousands of networks. Indeed, the stochasticity over the architectures inside the model itself is also of interest, in consideration of multiple aspects: (*i*) In the spirit of stochastic regularization, the introduced architecture variability helps to regularize deep models from co-adaptation and over-fitting, in a more structured and global style than standard stochastic regularizations applied on local feature maps or weights [35, 39, 19, 8]. (*ii*) The trained

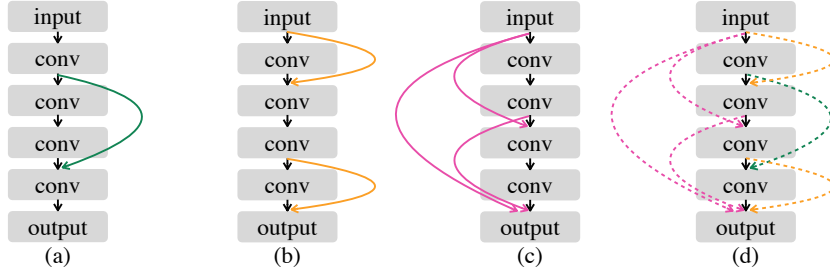

Figure 1: (Best viewed in color.) We plot four convolutions due to space limitation. (a)-(c): Diverse architectures lying in the wiring-based space. Note that (c) actually equals to the residual connections, and refer to [48] for the details of such equivalence. (d): The network with stochastic architecture is suitable for various architectures, and randomly activates an architecture at each training/test step. See Appendix A for practically used architectures.

weight-sharing network can adopt diverse architectures, seen or even unseen (as shown in Sec. 4.2) during training, to perform inference, enabling us to leverage the expressivity of various architectures with training only one set of weights. The predictions provided by different architectures can be further assembled or used to calculate uncertainty estimates, making the prediction model more *accurate*, *robust*, and *calibrated*.

In this work, we disentangle the Network with Stochastic Architectures (referred to as *NSA*) from the task of NAS, and provide a first systematical investigation on NSA as a stand-alone problem beyond NAS. At first, we investigate the un-identified characteristics and limitations of NSA. Through thorough empirical analyses, we have uncovered the *training/test disparity* and *mode collapse* issues of the vanilla NSA, which are non-trivial and neglected by existing works. We then develop several improvements to address these problems with theoretical and empirical insights. Furthermore, we also observe some remarkable features of NSA and our improved versions, such as good generalization capacity to unseen architectures, which could be intentionally leveraged to build NSA with enhanced predictive performance and hopefully benefit existing NAS methods. Finally, to fully exploit its potentials from inference-time stochastic architectures, we apply the NSA with our improvements into several challenging scenarios. Experimental results on multiple tasks testify the effectiveness of NSA. In summary, our contributions are as follows:

1. We provide a systematical investigation on the network with stochastic architectures (NSA).

2. We uncover a wide range of characteristics of NSA, identify two issues of it, *i.e.*, training/test disparity and mode collapse, and propose two techniques to address the issues.

3. We extend NSA into scenarios like *model ensemble*, *uncertainty estimation*, and *semi-supervised learning*, to enjoy the benefits from the stochasticity over the architectures. Extensive experiments prove the effectiveness of NSA in these scenarios.

## 2    NSA: Network with Stochastic Architectures

Before delving into the details of understanding and exploring NSA, we describe its basic definition and motivation, as well as its training and test principles. Then we briefly present its building details.

**What is NSA?** Basically, NSA is defined as a network with a fixed set of weights, but stochastically sampled architectures in both training and inference, distinct from the regular DNNs. To meet such a definition, the space, where we sample architectures, is usually required to be well structured [30, 22, 44], so that the shared weights can be architecture compatible. There are two popular structured architecture spaces: ($i$) in a sub-graph view – different architectures are different sub-graphs of a super graph with redundant computational branches [30, 22]; ($ii$) in a wiring view – different architectures activate different skip-connections among a fixed number of computational operations [44, 48] (see Fig. 1 for the details). In this work, we consider the latter as: ($i$) the operation redundancy in the former may probably limit the convergence of network weights; and ($ii$) the latter has a higher alignment with the classic ResNets [12] and DenseNets [14]. We parameterize the architecture as the discrete adjacency matrix of the directed graph on the fixed set of operation nodes.

**Why do we need a NSA?** At first, the architecture stochasticity is likely to regularize the training properly, in light of the stochastic regularization scheme. A more promising aspect is that given a trained NSA, we can evaluate the incoming data with diverse architectures thanks to the plug-and-play nature of the model for neural architectures. This enables us not only to evaluate a broad range of architectures with only the training efforts of once, but also to exploit the diverse predictive behaviors of different architectures, which are thought to carry specialized inductive bias. Moreover, the predictions from different architectures can further be assembled or integrated to calculate uncertainty estimates, giving rise to a more accurate, robust, and calibrated prediction model.

**Training principles.** To train a NSA to predict well under various architectures, we minimize the expected empirical risk *w.r.t.* the variable architecture for weight updating, as suggested in [1, 45, 3, 11]. Specifically, we assume that the architecture $\boldsymbol{\alpha}$ follows a distribution $p(\boldsymbol{\alpha})$, and we have access to a training set $\mathcal{D} = \{(\mathbf{x}_i, y_i)\}_{i=1}^n$ of size $n$, where $\mathbf{x}_i \in \mathbb{R}^d$ and $y_i \in \mathcal{Y}$ denote the data and label, respectively. The loss function for training is formulated as:

$$L(\mathbf{w}) = \mathbb{E}_{\boldsymbol{\alpha} \sim p(\boldsymbol{\alpha})} \Big[ \frac{1}{n} \sum_{(\mathbf{x}_i, y_i) \in \mathcal{D}} -\log p(y_i|\mathbf{x}_i; \mathbf{w}, \boldsymbol{\alpha}) \Big] \approx \frac{1}{|\mathcal{B}|} \sum_{(\mathbf{x}_i, y_i) \in \mathcal{B}} -\log p(y_i|\mathbf{x}_i; \mathbf{w}, \boldsymbol{\alpha}), \ \boldsymbol{\alpha} \sim p(\boldsymbol{\alpha}), \ (1)$$

where $\mathbf{w}$ denotes the weights, $\mathcal{B}$ represents a stochastic batch of data, and $p(y|\mathbf{x}_i; \mathbf{w}, \boldsymbol{\alpha})$ is the predictive distribution. Note that the sampled architecture is typically used for the whole batch [45, 3]. Given this, we can iteratively perform stochastic gradient descent (SGD) for weight training.[2]

**Test principles.** Based on a trained NSA, we can predict for the validation data with diverse architectures seen or even unseen (we will justify this in Sec. 4.2) during training, due to its high compatibility with various architectures. The accuracy on the validation data $\mathcal{D}_{\text{val}}$ of a specific architecture $\boldsymbol{\alpha}_0$ takes the form of $\mathcal{A}(\boldsymbol{\alpha}_0) = \frac{1}{|\mathcal{D}_{\text{val}}|} \sum_{(\mathbf{x}_i, y_i) \in \mathcal{D}_{\text{val}}} \mathbb{I}\big(\arg\max_y p(y|\mathbf{x}_i; \mathbf{w}, \boldsymbol{\alpha}_0) = y_i\big)$. Unlike regular DNNs, we can also ensemble the predictions from $T$ architectures $\{\boldsymbol{\alpha}_t\}_{t=1}^T$ for performance estimation: $\mathcal{A}_{\text{ens}} = \frac{1}{|\mathcal{D}_{\text{val}}|} \sum_{(\mathbf{x}_i, y_i) \in \mathcal{D}_{\text{val}}} \mathbb{I}\big(\arg\max_y \big(\frac{1}{T} \sum_{t=1}^T p(y|\mathbf{x}_i; \mathbf{w}, \boldsymbol{\alpha}_t)\big) = y_i\big)$. Note that NAS always takes $\mathcal{A}(\boldsymbol{\alpha}_0)$ as a proxy of $\boldsymbol{\alpha}_0$'s stand-alone performance (*i.e.*, the performance of the architecture with individual weights trained from scratch) to guide architecture search.

**A refined training space of architecture.** To avoid meaningless architecture samples, we adopt a knowledge guided sampler, *i.e.*, the Erdős-Rényi (ER) [6] model with $0.3$ probability to activate any one of the possible skip-connections, suggested by [44], to sample from the whole space. We also demand there is an overall chain-like connection. Before training, we refine the broad architecture space by randomly sampling a subset of it in the size of $S$ with the sampler. The $S$ architectures span the training space, and we uniformly choose one of them at each iteration during training (corresponding to $p(\boldsymbol{\alpha})$ in Eq. (1)). We apply such a refined training space of architectures for two reasons: (*i*) though the original architecture space is huge (*e.g.*, $> 10^{10}$), the architectures that could be sampled and used in training are limited owing to the limited training steps (*e.g.*, $< 10^5$); (*ii*) by fixing the training space at a certain size, we can further examine if training with more architectures harms the convergence of NSA and can the NSA trained under a limited number of architectures generalize to unseen architectures, as revealed in Sec. 3 and Sec. 4.2, respectively.

**Network details.** By convention, on the CIFAR-10 [16] task, the deployed network is divided into 3 stages in different spatial sizes, each containing 8 convolution modules, and we randomly sample an individual architecture for each stage. We use wide convolutions with the widening factor of 10 for feature extraction, inspired by their success in Wide Residual Networks (WRNs) [49]. A uniform sum precedes each convolution module, *i.e.*, a ReLU-Conv-BN triplet, to aggregate incoming feature maps. We use explicit down-sampling modules between stages to make the feature maps in every stage have the same size and hence can be freely connected. As a note, the residual connections also lie in the space (shown in Fig. 1(c)), allowing us to implement a comparable WRN as a baseline (denoted as WRN-28-10[†]). The training cost of NSA is almost identical to that of WRN-28-10[†], taking about 0.6 GPU day on a GTX 2080Ti for 300 training epochs.

## 3 Training/test Disparity of NSA

Though NSA has been widely deployed in NAS, the focus was mainly placed on evaluating the architecture candidates given a trained NSA, while leaving several important characteristics of NSA

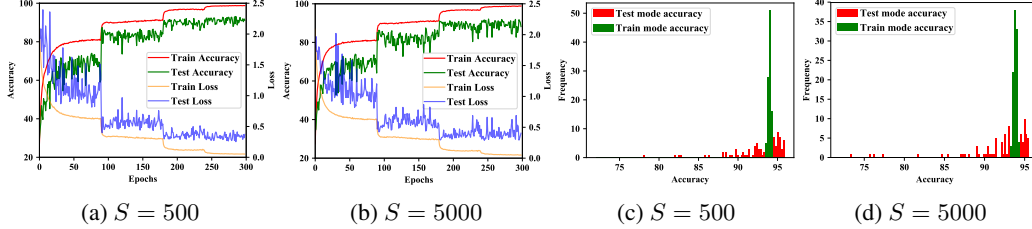

(a) $S = 500$  (b) $S = 5000$  (c) $S = 500$  (d) $S = 5000$

Figure 2: (a)-(b): The training curves of NSA trained under architecture spaces of size 500 and 5000. (c)-(d): The histograms for the validation accuracy of 100 random architectures (seen during training) with the training mode of BN turned on/off.

unexplored, such as the training stability, convergence, and sensitivity to the training architecture space size. In this section, we examine these previously ignored aspects of NSA and present the key observation of sharp training/test disparity. We further draw theoretical insights from the mechanism of Batch Normalization (BN) [15] to explain this phenomenon and propose solutions accordingly.

We start by examining NSA on the typical CIFAR-10 image classification task, with the number of used architectures $S$ during training varying from 500, to 5000 and 50000. We calculate the accuracy and loss of every batch of data given a batch-specific random architecture, according to Eq. (1), and take the average as the whole dataset accuracy and loss. We draw the training curves in Fig. 2(a)-(b) and Appendix B, respectively. Surprisingly, the training of NSA is stable and well converged in all cases, as well exhibiting tolerance to the variability of the training architecture space. This is somewhat counter-intuitive as there seems to be higher architecture variability in a wider space, rendering the data-fitting harder. We speculate that the well structured architecture space yields inherently consistent architecture samples, resulting in such results.

A much more attractive part is the test loss and test accuracy curves, owing to its severe instability. Typically, the training and test disparity of a DNN model is caused by the inconsistency inside BN – during training, batch specific statistics are used to normalize features while in test, their exponential moving average (ema) takes over to make the inference stable and behavior independent. To confirm this, we calculate the validation accuracy of 100 random architectures with the training mode of BN turned on or off, and plot their histograms in Fig. 2(c)-(d). The visualization echoes our speculation: the model behaviour becomes significantly unstable when replacing the training mode with the test one. This phenomenon is also found by some recent works [47, 46]. It may also explain why methods in NAS tend to evaluate the architectures with the training mode on [22, 45], though the validation results given by training-mode BN are not pretty reliable.

To figure out the underlying reasons of such a problem, we first of all draw some insights from the formulation of BN. Given a mini-batch of $|\mathcal{B}|$ instances, we consider a single channel of the batch features $\{h_{1,\boldsymbol{\alpha}}, h_{2,\boldsymbol{\alpha}}, ..., h_{|\mathcal{B}|,\boldsymbol{\alpha}}\}$, with the assumption that the spatial dimension is 1 for simplicity. $\boldsymbol{\alpha}$ in the subscript refers to the used architecture for the batch. BN works by applying the following transformation on the features (the affine transformation is omitted):

$$\mu = \frac{1}{|\mathcal{B}|}\sum_{i=1}^{|\mathcal{B}|} h_{i,\boldsymbol{\alpha}}, \ \ \sigma^2 = \frac{1}{|\mathcal{B}|}\sum_{i=1}^{|\mathcal{B}|}(h_{i,\boldsymbol{\alpha}} - \mu)^2, \ \ \hat{h}_{i,\boldsymbol{\alpha}}^{\text{train}} = \frac{h_{i,\boldsymbol{\alpha}} - \mu}{\sqrt{\sigma^2 + \epsilon}}, \ \ \hat{h}_{i,\boldsymbol{\alpha}}^{\text{test}} = \frac{h_{i,\boldsymbol{\alpha}} - \mu_{\text{ema}}}{\sqrt{\sigma_{\text{ema}}^2 + \epsilon}}. \quad (2)$$

Then we look at the variance of $\mu$, since that the gap between $\mu$ and the constant $\mu_{\text{ema}}$ is the major discrepancy between training and test, and obtain: $\text{var}(\mu) = \frac{1}{|\mathcal{B}|^2}(\sum_{i=1}^{|\mathcal{B}|}\text{var}(h_{i,\boldsymbol{\alpha}}) + \sum_{i\neq j}\text{cov}(h_{i,\boldsymbol{\alpha}}, h_{j,\boldsymbol{\alpha}}))$. Intuitively, the features generated with the same architecture $\boldsymbol{\alpha}$ are highly correlated, because the architecture commonly shifts the features toward a certain direction. This makes the second term of the decomposition of $\text{var}(\mu)$ undesirably large, given that it is a summation over $|\mathcal{B}| \times (|\mathcal{B}| - 1)$ terms. So, NSA uses batch statistics varying across architectures during training, but uses architecture agnostic ones during test, bringing inconsistency and hence unstable prediction.

With the root of the problem diagnosed, we gain the opportunities to solve it. To reduce the correlation between the features in a batch, a straight-forward solution is to reduce the correlation between the architectures which generate these features. Continuing from this, we replace the batch specific

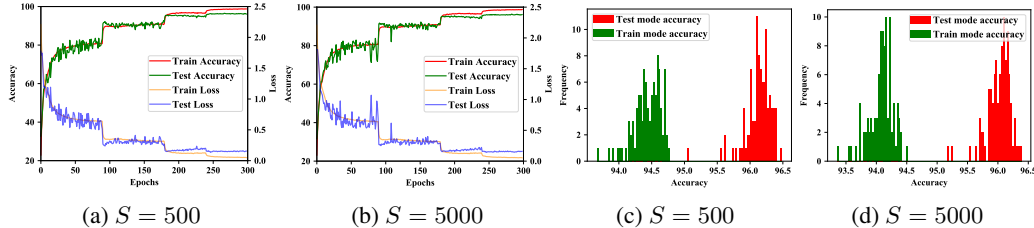

$$\text{(a) } S = 500 \qquad \text{(b) } S = 5000 \qquad \text{(c) } S = 500 \qquad \text{(d) } S = 5000$$

Figure 3: (a)-(b): The training curves of NSA-i trained under architecture spaces of size 500 and 5000. (c)-(d): The histograms for the validation accuracy of 100 random architectures tested upon the trained NSA-i, with the training mode of BN turned on/off.

architectures in Eq. (1) with instance specific ones[3]:

$$L^*(\mathbf{w}) = \frac{1}{|\mathcal{B}|} \sum_{(\mathbf{x}_i, y_i) \in \mathcal{B}} - \log p(y_i | \mathbf{x}_i; \mathbf{w}, \boldsymbol{\alpha}_i), \quad \boldsymbol{\alpha}_i \sim p(\boldsymbol{\alpha}), i = 1, ..., |\mathcal{B}|. \tag{3}$$

Then, the batch mean is re-calculated as $\mu^* = \frac{1}{|\mathcal{B}|} \sum_{i=1}^{|\mathcal{B}|} h_{i,\boldsymbol{\alpha}_i}$, with variance $\text{var}(\mu^*) = \frac{1}{|\mathcal{B}|^2} (\sum_{i=1}^{|\mathcal{B}|} \text{var}(h_{i,\boldsymbol{\alpha}_i}) + \sum_{i \neq j} \text{cov}(h_{i,\boldsymbol{\alpha}_i}, h_{j,\boldsymbol{\alpha}_j}))$. $\text{cov}(h_{i,\boldsymbol{\alpha}_i}, h_{j,\boldsymbol{\alpha}_j})$ should be small given that the architecture for each data is *i.i.d.*, thus ideally we can reduce the variance of the batch mean by one order of magnitude. We refer to network trained with Eq. (3) as improved NSA (NSA-i), and take NSA-i as the default model in the following evaluation for its advantages.

We provide the training curves and validation results of NSA-i in Fig. 3. As expected, in Fig. 3(a)-(b), the test results are much more stable and consistent with the training ones. Surprisingly, in Fig. 3(c)-(d), test-mode BN induces notably better validation accuracy, implying the weaknesses of training-mode BN: the training statistics commonly cannot approximate the whole dataset ones well. A direct comparison on $\text{var}(\mu)$ between NSA and NSA-i is deferred to Appendix C.

At last, we have done another interesting study – ranking 100 random architectures *w.r.t.* their validation accuracy with the training mode of BN turned on or off, and calculating the Spearman rank correlation [27] between the two modes (the higher, the more correlated). The results of NSA are 0.33, 0.37, and 0.258 with the training space containing 500, 5000, and 50000 architectures, respectively. As a comparison, NSA-i offers 0.588, 0.395, and 0.615. This testifies that the architecture assessment provided by NSA, trained with batch specific architectures, is indeed less stable than that from NSA-i. This also highlights the necessity of solving the BN problem in NAS (either with the investigated architecture space or with the popular DARTS space [22]), and challenges the effectiveness of using training-mode BN for architecture evaluation, as in almost all efficient NAS methods.

## 4 Inference-time Properties of NSA

In this section, we use the trained NSA for inference, and aim to analyze some properties of its inference-time behaviour. We concern (*i*) Do diverse architectures behave diversely given shared weights? (*ii*) Can NSA trained under a limited architecture space generalize to unseen architectures in the broad, raw architecture space? The two aspects are of central importance for both architecture evaluation and ensemble with various architectures. We answer the two questions in the following.

### 4.1 Mode Collapse of Diverse Architectures

As stated, the network architecture is capable to carry specific inductive bias, thus different architectures may deliver diverse predictions for the same data. Such predictive diversity is comprehensively helpful to the model, ranging from enhancing performance [20] and robustness [29], to yielding more calibrated uncertainty estimates [41]. But can the predictive diversity still be held given only a set of shared weights in NSA? Intuitively, the answer is not positive, because the weights in NSA are architecture agnostic to permit the trained weights generalize across the whole architecture space. The

expectation *w.r.t.* architecture in the training loss forces the weights to be robust against architecture variability, and the model to predict consistently under diverse architectures. Thus, the network would seemingly yield architecture agnostic prediction, referred to as the *function mode collapse*, and lose the advantages of exploring diverse modes of prediction behaviour from multiple architectures.

Based on these speculations, we launch a set of experiments to identify whether mode collapse indeed exists or not. A realistic barrier is that the prediction behaviour of a network model can hardly be numerically measured, owing to its black-box nature. Drawing inspiration from the fact that model ensemble frequently benefits from diverse base predictors [20], we opt to use the ensemble performance gain as a metric, to estimate the behaviour diversity of different architectures. Specifically, we test on the NSA-i model, given its supremacy over naive NSA, trained with $S = 500$ architectures. We let the ensemble number of architectures $T$ range from 1 to 500, and draw the change of ensemble accuracy *w.r.t.* $T$ in Fig. 4.

As shown, the ensemble performance gain is limited (almost 0.003) and stops increasing quickly. Such results substantiate that there are moderate levels of function mode collapse among various architectures when using a shared set of weights.

We know the source of this issue is the shared weights are architecture agnostic, then as a solution, we can augment the shared weights with an extra set of architecture dependent weights, to enjoy the benefits from more diverse function modes of different architectures[4]. The extra weights of every architecture should be low-dimensional, because at per training step, only the extra weights of several architectures (no more than batch size considering Eq. (3)) would be updated. If not, they will not be trained thoroughly. Under this consideration, we employ architecture dependent aggregation and BN in NSA-id, following the style of

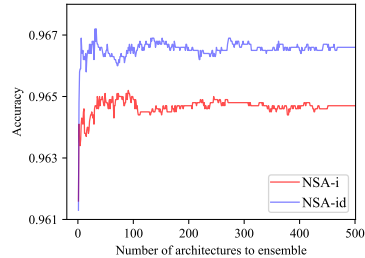

Figure 4: The change of the ensemble performance *w.r.t.* the number of architecture used to ensemble.

the class-conditional BN widely used in conditional generative modeling [26]. Namely, we build an individual set of trainable aggregation coefficients and BN affine parameters for each architecture, and select the corresponding set to the architecture for calculation at per step. We refer to NSA-i with architecture dependent weights as NSA-id.

Then, we assess the mode collapse level of NSA-id with the aforementioned ensemble based evaluation. To compare with NSA-i fairly, we use only the architecture conditional aggregations, which introduces negligible extra weights, in this experiment. We exhibit the results in Fig. 4. As expected, the ensemble gain is more obvious compared to NSA-i[5]. However, identical to NSA-i, NSA-id cannot enjoy further ensemble gain after seeing almost 20 architectures. We think it is reasonable: as discussed, the main weights of the network are architecture agnostic, rendering it hard to exhaustively diversify the predictions of various architectures with only few additional weights. To summarize, mode collapse indeed occurs and employing architecture conditional weights mitigates it.

## 4.2 Generalization Capacity to Unseen Architectures

As we stated, the inference-time architecture stochasticity of NSA is desirable, making us capable of exploiting the predictive power of various architectures. But does the trained NSA only accommodate the architectures seen during training? Can the trained NSA generalize to unseen architectures for broader exploration? Here, we offer answers for them with both qualitative and quantitative evidence.

First, we calculate the test accuracy of 200 randomly sampled architectures based on the NSA-i models trained under various spaces (as shown in Appendix D, the naive NSA models with training-mode BN would provide similar results). A half of the 200 architectures are seen during training while the other half not. We depict the test accuracy histograms of the two types of architectures in Fig. 5. An intuitive conclusion could be drawn is that with the training architecture space large enough (*i.e.*, $S \geq 500$), the trained NSA-i can present matched performance on the unseen architectures with

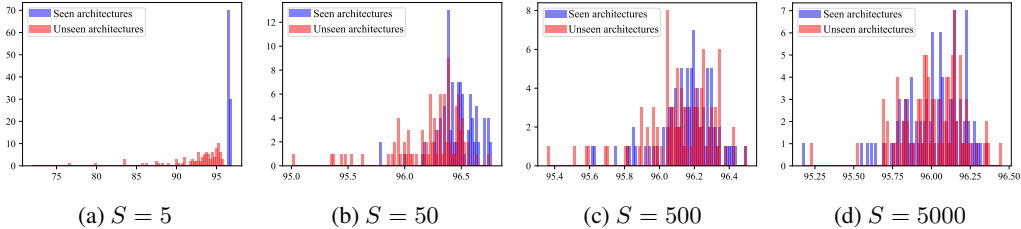

(a) $S = 5$      (b) $S = 50$      (c) $S = 500$      (d) $S = 5000$

Figure 5: The histograms for the validation accuracy of 100 architectures seen during training vs. those for 100 unseen architectures, tested on the trained NSA-i models with different training space sizes.

the ones used for training. When the training space is too narrow (*e.g.*, $S \leq 50$), the network behaves distinctly over the two classes of architectures, ruling out the generalization across architectures.

An alternative to quantitatively analyze the generalization capacity of NSA-i is to check whether we can distinguish the seen architectures from the unseen ones *w.r.t.* their validation accuracy. A golden metric to estimate the goodness of a classifier on such a binary classification task is the *Area under the ROC Curve* (AUC). Thus, we report the AUCs of the trained NSA-i models in Table 1. We also report the average accuracy of the seen architectures and the unseen ones for reference. Consistent with the histograms, with the training space increases, it is harder to differentiate these two classes of architectures *w.r.t.* validation accuracy; when $S \geq 500$, almost any binary classifier randomly guesses, given the near 0.5 AUCs. Meanwhile, we also notice that the average validation accuracy of seen architectures decreases slightly.

Table 1: The change of the AUC, which measures the differentiability between the seen architectures and unseen ones given the validation accuracy, *w.r.t.* the training space size $S$. We also report the average accuracy of the seen and unseen architecture for reference.

| $S$ | AUC | Avg acc. (seen) | Avg acc. (unseen) |
|---|---|---|---|
| 5 | 1.00 | 96.57% | 91.81% |
| 50 | 0.77 | 96.47% | 96.23% |
| 500 | 0.57 | 96.16% | 96.12% |
| 5000 | 0.52 | 96.01% | 96.01% |

These results validate the generalization capacity of NSA, perhaps because the shared weights learn common structures of the architectures. As shown, we can train a NSA with a suitable number of architectures (*e.g.*, $[500, 5000]$) to conjoin architecture generalization and accuracy. We hope that this may also serve as an insightful heuristic for training weight-sharing proxy networks in NAS.

## 5 Applications of NSA

Given the potential of NSA to unleash the predictive capacity of diverse architectures during inference, in this section, we apply NSA to a variety of tasks ranging from ensemble learning, uncertainty estimation, to semi-supervised learning, which is unexplored in previously works. As discussed, we take WRN-28-10[†] as a main baseline, for its identical settings with NSA and the strong competitiveness of residual connections [12, 49]. As a note, NSA leverages stochastic architectures for inference, consistent with the empirical Bayes methods, *e.g.*, Monte Carlo (MC) dropout [7]. Thus, we take MC dropout built upon WRN-28-10[†] as another baseline.

**Hyper-parameter setting.** We enable the architecture conditional BNs in the following experiments. We empirically found that using more than 10 architectures in training frequently results in worse validation results, possibly due to the incomplete training of the redundant weights in BNs, as explained in Sec. 4.1. Therefore, we use $S = 5$ randomly sampled architectures for training and inference. To further facilitate the convergence of the shared weights, we deploy an auxiliary classifier following [22] with 0.1 loss coefficient (also deployed in the baselines). We apply standard data processing and CutOut augmentation [5]. The optimization settings follow WRN-28-10 [49].

### 5.1 Model Ensemble with Stochastic Architectures on CIFAR-10 and CIFAR-100

As shown in Sec. 4.1, there is evidence to suggest that ensemble the predictions from different architectures does boost the validation performance, consistent with the common knowledge [18], so we continue evaluating this technique on the more expressive NSA-id models with conditional BNs used. For the results of NSA-id, as stated, we use $S = 5$ architectures for training and ensemble

Table 2: Comparison of NSA-id, using ensemble of 5 different architectures for prediction, and a range of competing baseline, in terms of test error and ECE. ENAS and DARTS adpot the parameter-efficient separable convolutions and apply *re-training* to get the results.

| Method | # params | CIFAR-10 | | CIFAR-100 | |
|---|---|---|---|---|---|
| | | Test error (%) $\downarrow$ | ECE $\downarrow$ | Test error (%) $\downarrow$ | ECE $\downarrow$ |
| WRN-28-10 [49] | 36.5M | 4.00 | - | 19.25 | - |
| DenseNet-BC [14] | 25.6M | 3.46 | - | 17.18 | - |
| ENAS + CutOut [30] | 4.6M | 2.89 | - | - | - |
| DARTS + CutOut [22] | 3.4M | 2.83 | - | - | - |
| WRN-28-10$^{\dagger}$ | 39.5M | 2.93 | 0.0140 | 16.75 | 0.0672 |
| WRN-28-10$^{\dagger}$, MC dropout | 39.5M | 3.23 | 0.0107 | 17.16 | 0.0454 |
| Average of individuals | 39.5M | 2.97 | 0.0153 | 17.02 | 0.0446 |
| NSA-id | 39.6M | **2.75** | **0.0032** | **16.44** | **0.0212** |

all of them for test (*i.e.*, $T = 5$). For WRN-28-10$^{\dagger}$ with MC dropout, we predict one data for 100 times with randomly sampled dropout masks and assemble them. We implement a further baseline: *Average of individuals*, in which we individually trains 5 networks with the 5 architectures used by NSA-id, and report their average results, to present the average performance of the used architectures, rather than their ensemble as comparing that to NSA-id is unfair given the need of $5\times$ training costs.

We report the results in Table 2. In both tasks, NSA-id surpasses the strong baselines with clear margins. The 2.75% error rate on CIFAR-10 is rather promising considering the wide convolution based backbone. Notably, the comparison between NSA-id and *Average of individuals* confirms that ensembling multiple architectures leads to improved performance over a single architecture, despite using shared weights[6]. These results also prove the ER-0.3 model provides a good architecture space.

We also detail the model calibration, which is another major concern of classification model, in Table 2. Following [10], we take expected calibration error (ECE) as a measure of calibration. Surprisingly, NSA-id shows lower ECE than the strong, principled baseline MC dropout with huge margins. We think this results from the fact that different architectures offer relatively diverse predictions in NSA-id, alleviating the over-confidence, while MC dropout is known to suffer from mode collapse [18], and hence cannot benefit too much from prediction ensemble.

## 5.2 Uncertainty Estimation

In NSA-id, the architecture stochasticity results in the predictive uncertainty, permitting us to regard NSA-id as an empirical Bayes method. In this section, we assess the uncertain estimates provided by NSA-id. Suggested by [34], we adopt the mutual information (MI) between the prediction of incoming data and the model parameters as the uncertainty measure, namely, $\mathcal{I}(\mathbf{w}, \boldsymbol{\alpha}, y | \mathcal{D}, x) \approx H[\frac{1}{T}\sum_{i=1}^{T} p(y|x; \mathbf{w}, \boldsymbol{\alpha}_i)] - \frac{1}{T}\sum_{i=1}^{T} H[p(y|x; \mathbf{w}, \boldsymbol{\alpha}_i)]$, where $H$ is the entropy, and $T = 5$ as stated. We test the uncertainty estimates on two challenging kinds of samples: out-of-distribution (OOD) ones and adversarial ones. The models to evaluate are trained on CIFAR-10, and OOD samples refer to the test data of SVHN. For the adversarial samples, we use the frequently adopted, performant Projected Gradient Descent (PGD) [25] to craft. In practice, we first calculate the MI of normal test samples and OOD (or adversarial) ones, then we compute and report the AUC of the binary classification of directly distinguishing the normal ones (class 0) from OOD (or adversarial) ones (class 1) based on the MI. The underlying notion is that OOD (or adversarial) samples commonly deviate from the manifold of normal ones, thus have high uncertainty. As shown in Table 3, NSA-id consistently displays improved uncertainty estimates than the golden baseline WRN-28-10$^{\dagger}$ with MC dropout. Also of note that, NSA-id shows stronger adversarial robustness against PGD attack.

## 5.3 Semi-supervised Learning

At last, we use NSA-id to perform semi-supervised classification on CIFAR-10, using only 4000 labeled data. The notion of applying NSA-id into such a scenario is that with the uncertainty estimates provided by NSA-id, we can minimize the predictive uncertainty (*i.e.*, the aforementioned mutual information) of unlabeled data, to assist the learning with the labeled data. The uncertainty minimization

Table 3: Comparison between NSA-id and MC dropout in terms of the quality of uncertainty estimates. PGD$a$-$b$-$c$ represents the PGD adversary with perturbation budget $a/255$, number of steps $b$, and step size $c/255$.

| Method | OOD | PGD1-2-1 | | PGD2-3-1 | | PGD3-4-1 | |
|---|---|---|---|---|---|---|---|
| | AUC ↑ | Acc. ↑ | AUC ↑ | Acc. ↑ | AUC ↑ | Acc. ↑ | AUC ↑ |
| WRN-28-10[†], MC dropout | 0.935 | 0.622 | 0.735 | 0.345 | 0.694 | 0.183 | 0.564 |
| NSA-id | **0.970** | **0.630** | **0.737** | **0.401** | **0.705** | **0.263** | **0.618** |

is achieved by optimizing a consistency loss: $L_{\text{unlabeled}} = \sum_i ||p(y|x_i^{un}; \mathbf{w}, \boldsymbol{\alpha}_i) - p(y|x_i^{un}; \mathbf{w}, \boldsymbol{\alpha}_i')||_2^2$, where $x_i^{un}$ denotes the $i$th unlabeled data, and $\boldsymbol{\alpha}_i$ and $\boldsymbol{\alpha}_i'$ are two randomly sampled architectures. In practice, we use the distance between output logits instead of that of probabilities. We set the coefficient of consistency loss to be 20, following an anneal schedule [17]. After training, we assemble the predictions of different architectures for final prediction. We implement two baselines: ($i$) WRN-28-10[†] with $\Pi$ model [17], which works the same as NSA-id expect for using dropout to provide twice predictions for one data; ($ii$) WRN-28-10[†] trained with only labeled data. The validation accuracy of NSA-id and the two baselines are 86.96%, 85.22%, and 83.87%, respectively. NSA-id outperforms $\Pi$ model, perhaps because the architecture stochasticity can explore more diverse predictions across decision boundaries, thus penalizing the inconsistency between the predictions of unlabeled data would drive the decision boundaries to be more robust.

# 6 Related Work

Randomizing certain parts of DNNs is usually indispensable to prevent the trained model from over-fitting, over-confident, and co-adaptation [35, 39, 19, 8, 21]. But existing stochastic regularizations are commonly applied locally upon the network weights or the hidden feature maps, which is argued to be less effective than globally regularizing the behaviour of the model [4], as done in NSA. Besides, these stochastic regularizations are usually turned off in inference phase, while NSA predicts with stochastic architectures and benefits from such stochasticity. A more principled approach to include stochasticity is Bayesian Neural Networks (BNNs) [24, 28, 9, 2, 23, 7], which place uncertainty upon network weights and measure predictive uncertainty given Bayesian theorem. But BNNs are known to suffer from mode collapse [18] and training challenges [50], hence not popularly used in practice.

In neural architecture search (NAS) [52, 53, 31, 30, 22, 45, 3, 44, 37], tremendous efforts have been devoted to discovering performant architectures in a broad yet structured architecture space. For computationally feasible search, it is common to train a network with stochastic architectures to enable the evaluation of ample architectures given shared weights. But almost all NAS works ignore to analyze the properties of such a network, *e.g.*, the convergence, training stability, and generalization to unseen architectures, which are of central importance in NAS. In this work, we uncover these un-identified aspects, and provide novel insights for how to train a better NSA in NAS.

# 7 Conclusion

In this work, we aim at understanding the properties of the network with stochastic architectures, and applying such a network in more extensive and suitable scenarios. Firstly, we reveal un-identified training and test properties of NSA. We observe two issues, training/test disparity and mode collapse, of NSA, which are ignored by previous works, and propose two novel approaches to address them. We further provide valuable insights on how to train a NSA, hopefully benefiting NAS. At last, we apply NSA into three appropriate scenarios to sufficiently exploit its potential and see good results.

## Broader Impact

This work manages to understand a wide range of properties of the network with stochastic architectures (NSA), and apply it to several challenging tasks to sufficiently exploit its potential. It has the following positive impacts in the society. First, as NSA is a broadly used technique in neural architecture search (NAS), our analyses can provide valuable insights for the following research on NAS. Second, the proposed improvements upon NSA are practically applicable, and NSA with such improvements has shown promise in various scenarios such as ensemble learning and semi-supervised learning. Third, as the architecture stochasticity can be leveraged to provide uncertainty estimates during inference, NSA has a potential to be used in practical applications where the uncertainty measures are crucial, such as finance, automatic driving, etc. At the same time, this work may have some possible negative consequences. For example, just like other NAS works, it would enable searching better neural architectures automatically, which may potentially result in job loss of many researchers and engineers in the future.

## Acknowledge

This work was supported by the National Key Research and Development Program of China (No.2017YFA0700904), NSFC Projects (Nos. 61620106010, U19B2034, U1811461), Beijing Academy of Artificial Intelligence (BAAI), Tsinghua-Huawei Joint Research Program, a grant from Tsinghua Institute for Guo Qiang, Tiangong Institute for Intelligent Computing, and the NVIDIA NVAIL Program with GPU/DGX Acceleration.

## Footnotes

[1]In NAS, the distribution where the stochastic architectures are sampled may also be updated *w.r.t.* validation data simultaneously, with the purpose of discovering outperforming architectures.

[2]We view $p(\boldsymbol{\alpha})$ as fixed for simplicity despite updating it *w.r.t.* validation results at the same time is feasible.

[3]Using instance specific architectures is feasible to implement when using the architecture space of [44], as done in this work, but is not directly implementable in the sub-graph based space, left as a future work.

[4]Of course, introducing architecture dependent weights will hinder the trained weights from generalizing to unseen architectures as we can only deploy extra weights for architectures seen during training.

[5]The performance drop in Fig. 4 may stem from the facts that the 500 used architectures are randomly sampled and we perform only uniform ensemble instead of weighted ensemble. Thus assembling more base learners may not give rise to rigidly better predictions.

[6]As a note, the ensemble of the 5 aforementioned *individuals* yields striking 2.36% error rate on CIFAR-10, confirming that weight sharing is a main cause of mode collapse.

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
