[Supplementary Material]

# Supplementary Material for:
# Understanding and Exploring the Network with Stochastic Architectures

**Zhijie Deng, Yinpeng Dong, Shifeng Zhang, Jun Zhu**[*]
Dept. of Comp. Sci. & Tech., Institute for AI, BNRist Center
Tsinghua-Bosch Joint ML Center, THBI Lab, Tsinghua University, Beijing, 100084 China
{dzj17,dyp17,zhangsf15}@mails.tsinghua.edu.cn, dcszj@mail.tsinghua.edu.cn

## A  The Practically Used Architectures in NSA-id

In this section, we plot the 5 randomly sampled architectures used in NSA-id in Sec. 5. As shown in Fig. 5, the 5 architectures are distinct from each other. Every architecture has diverse sub-architectures in the various stages. Based on the NSA framework, we can learn a set of network weights that are compatible with these different architectures, and the learned network can be applied into scenarios like model ensemble, uncertainty estimation, and semi-supervised learning.

## B  More Results of Vanilla NSA and NSA-i

We provide more results for the training and test behaviour of vanilla NSA and NSA-i in this section.

We draw the learning curves and validation histograms of vanilla NSA with the training architecture space containing 50000 samples (*i.e.*, $S = 50000$) in Fig. 6. Consistent with the results in Fig. 2, the training is converged and the large training space does not lead to significantly worse training accuracy and test one. Meanwhile, the test curves are pretty unstable, and the training-mode validation results are highly inconsistent with test-mode ones.

We display the corresponding results of NSA-i in Fig. 7, which provide substantial contrasts to vanilla NSA. The comparisons between these two figures further confirm the training/test disparity issue of vanilla NSA, and substantiate the effectiveness of NSA-i.

## C  Track the Values of var$(\mu)$

We take 1000 training images from CIFAR-10 as a fixed batch, randomly sample the neural architecture for inference, and compute var$(\mu)$ of the last BN layer of a NSA and a NSA-i trained given $S = 5000$ architectures. We calculate the average variance over all the channels and spatial locations, and the results of NSA and NSA-i are 0.00214 and 0.00082, respectively, which testify the effectiveness of NSA-i.

## D  Generalization Capacity to Unseen Architectures of Vanilla NSA

In this section, we calculate the test accuracy of 200 randomly sampled architectures based on the vanilla NSA models trained under various spaces. A half of these architectures are seen during training while the other half not. If training space size $S < 100$, namely, $S = 5$ or $S = 50$, the seen architectures will be uniformly re-sampled to build a set of seen architectures with 100 elements. As suggested by the results in Fig. 2 and Fig. 6, we use the training-mode BN during test. We plot the test accuracy histograms of the two types of architectures in Fig. 8. As shown, the results are similar with those in Fig. 5, and can also induce the conclusion in Sec 4.2.

---

[*]Corresponding author.

Figure 5: Five randomly sampled architectures used in NSA-id in Sec. 5.

# E  Extra Details about the Semi-supervised Learning Experiments

In the semi-supervised learning, we train NSA-id and the two baselines for total 100 epochs, with the learning rate decaying by 0.2 at 40th and 80th epoch. For NSA-id, we assemble the outputs of all the 5 architectures for prediction. For WRN-28-10[†] with Π model, we predict one data for 100 times with randomly sampled dropout masks and assemble them. We also adopt early stopping trick to mitigate the over-fitting on limited labelled data.

Figure 6: Left: training curves of vanilla NSA. Right: histograms for the validation accuracy of 100 random architectures with the training mode of BN turned on/off (calculated given a trained vanilla NSA). The training architecture space consists of 50000 samples.

Figure 7: Left: training curves of NSA-i. Right: histograms for the validation accuracy of 100 random architectures with the training mode of BN turned on/off (calculated given a trained NSA-i). The training architecture space consists of 50000 samples.

(a) $S = 5$    (b) $S = 50$    (c) $S = 500$    (d) $S = 5000$

Figure 8: The histograms for the validation accuracy of 100 architectures seen during training vs. those for 100 unseen architectures, tested on the trained vanilla NSA models with training-mode BN.