[Reviews · NeurIPS 2020]

Review 1

Summary and Contributions: The paper investigates the role of network with stochastic architectures, and identifies two issues associated with it: training/test disparity and mode collapse. Correspondingly, the authors propose their solutions to alleviate these issues. Experimental results demonstrate the effectiveness of their techniques, and also show the promising extensions of NSA in various applications. Overall it is novel and interesting to explore these problems, despite that some analysis are not clear enough to illustrate the problem and solutions. [The authors feedback mostly addresses my concerns and I am in favour of acceptance.]

Strengths: 1. The paper investigate a number of interesting and promising points in neural architecture search, e.g., train/test disparity, mode collapse, architecture generalization, which are seldom explored in previous work. 2. The authors conduct a set of analysis to support the claims, and experiments are adequate and informative. 3. Various extensions of NSA are provided, demonstrating the superiority of NSA in specific scenarios.

Weaknesses: 1. The paper seeks to answer a set of important research questions, however, for some questions, the analysis is a bit complicated and not very convincing. See the detailed comments below. 2. Throughout the analysis, there are some sections that establish the solution without checking the premise or assumptions first. See the detailed comments below. 3. The writing is a bit hard to follow. See the comments in [Clarity].

Correctness: Most analysis are based on intuition, verified by empirical results. It is hard to say scientifically correct, but could be empirically effective.

Clarity: The overall structure is clear but the writing is a bit hard to follow in the some parts. For instance: 1) in L212-L216, it is hard to follow the exact solution. It would be better to add some formulas or diagrams, even in the Appendix. 2) In L295, how is the mutual information used in that experiment?

Relation to Prior Work: The relevant literature are adequately incorporated.

Reproducibility: Yes

Additional Feedback: 1. Have the authors tried to track the values of var(\mu) with NSA and NSA-i to testify the assumptions in the improvement: cov(h_{i,\alpha}, h_{j,\alpha}) are of high correlation while cov(h_{i,\alpha_i}, h_{j,\alpha_j}) are i.i.d.. This would be more direct to justify the proposed NSA-i. 2. In Sec 4.1, while the proposed NSA-id shows higher ensemble accuracies, it seems as if the models are still not diverse as the rise of accuracy saturate quickly. In other words, it is still questionable that the rise of accuracy is from the diversity of architectures. 3. In L206: how do we know weight sharing is the root cause of mode collapse. Are there any more formal descriptions to explain the cause? 4. The paper focus on a refined search space grounded on Wide-ResNet, which could be very different from the common search space employed in ENAS/DARTS. So far the empirical findings only hold for NSA, but not the general NAS. 5. How long does the searching algorithm take for training? I find a lot of connections among different layers of Wide-ResNet in the visualization of the Appendix, which could be very time-consuming. 6. In table 2, NSA uses significantly more parameters (and probably more computational FLOPs) comparing to DARTS and ENAS, and thus the slight improvement of accuracy is less surprising.


Review 2

Summary and Contributions: The authors investigate the source of multiple issues in neural architecture search (NAS) by drilling down on problems with the underlying neural stochastic architectures (NSA). Specifically, the authors identify batch norm as the root cause in the observed train-test disparity in NSA. Batch norm typically has different methods to calculate batch statistics at train vs test time and this causes test-time predictions of NSA to be much more stochastic and lower performing than train time predictions. Additionally, the authors look at “mode collapse” of model weights in NSA. This refers to how NSA weights naively would converge on weights that are robust to different architectures rather than weights that support diverse predictions. The authors solve this issue by augmenting NSA models with architecture specific weights. Finally, the authors investigate how training on a limited subset of the architecture space affects performance when NSA weights are applied to previously unseen architectures. The authors report that when NSA-i models are trained on “enough” (~500) architectures they generalize fine to unseen architectures.

Strengths: -Strong experiments that demonstrate the claims of the authors -Fig 2 vs Fig 3 is a compelling demonstration of how changing batch norm properties can improve train-test performance -Good baselines in Table 2 -Batch norm implementation over the architectures seems like a great step for NSA

Weaknesses: I would be interested to see how NSA performs on regression tasks. The authors focus on classification tasks alone. In regression tasks, one can investigate the variance of the predictions over many architectures as a metric of uncertainty quantification. In classification, there's no equivalent metric since one typically does not consider the variance of p(y_i=K|x_i) for a particular class because the "variance" of a probability is a non-standard notion.

Correctness: The claims all appear to be correct to me. I checked all the equations and found no errors and the empirical methodology is sound.

Clarity: Yes. The authors succinctly explain their concepts and present it in an approachable way. I noticed no clunky phrases or grammatical issues/typos. Well done.

Relation to Prior Work: Yes, with the caveat that the author missed a reference to a related piece of work from NeurIPS 2019 https://arxiv.org/pdf/1812.09584.pdf

Reproducibility: Yes

Additional Feedback: Updating my review to say that I acknowledge the author feedback and I've updated my score by a point because of the thoughtfulness of the feedback.


Review 3

Summary and Contributions: This work explores the properties of the network with stochastic architectures (NSA). They propose some solutions to issues found and apply the NSA to model ensembling, uncertainty estimation and semi-supervised learning. They show in their evaluation that their proposal improve performance on the application mentioned before.

Strengths: * The paper is well-written, well-structured and easy to understand. I really enjoyed reading this paper. * The paper identified two problems and proposes solutions that empirically improves the optimization. * The paper presents diverse evaluation and improve on state-of-the-art classification, uncertainty estimation and semi-supervised learning ==== Post-rebuttal ==== Thank you to the authors for the rebuttal and clarifying some open questions. I found this paper really interesting and therefore, keep my score and vote in favour of accepting it.

Weaknesses: * The paper is mostly empirically based, and a major part focuses on the drawbacks of batch normalisation. * Conclusion is a more of a summary. The authors could have discussed more about what could have been investigate in the future.

Correctness: The claims and empirical methodology are adequate.

Clarity: I do believe, the paper is well-structured and well-written, thus making it easy to understand.

Relation to Prior Work: I do believe, the authors described clearly how his work differs from previous contributions.

Reproducibility: Yes

Additional Feedback: I have only some further questions: * l.81: What are typical choices for the distribution for alpha and how does this choice affect the optimization? * Eq.3 and Eq.1 look exactly the same to me, what is the difference? In the case of instance specific architecture, shouldn't the loss be calculated over the instance? * Figure 4: Shouldn't the dependent weights improve overall performance as claimed in the paper? I found it surprising that it decreases in performance. I do not particularly agree that the "ensemble gain is more obvious compared to NSA-i" as the plot does not clearly show it. Can you quantify this?

[Author Response · NeurIPS 2020]

We thank all reviewers for their constructive comments. We are encouraged that all voted to accept, and the acknowl-
edgement of the importance of our work [R1, R2] and the comprehensiveness of our studies [R1, R2, R3]. We address
specific comments below and will incorporate them to the updated version.

**To R1: Q: Writing.** Sorry for the hardness to follow in L212-L216. Recall that the basic cell in NSA is Aggregation-
ReLU-Conv-BN (see L111). Particularly, the aggregation module is to combine the input data from multiple edges
via a weighted sum (see Figure 5 in Appendix). To avoid introducing unaffordable architecture dependent parameters,
we employ architecture dependent aggregation and BN in NSA-id, following the style of the class-conditional BN
widely used in conditional generative modeling [*1]. Namely, we build an individual set of trainable aggregation
coefficients and BN affine parameters for each architecture. We'll rewrite this part and add a figure to depict this
explicitly. Regarding the mutual information (MI), a standard measure of uncertainty [33], we first calculate the MI of
normal test samples and OOD (or adversarial) ones, based on which we directly distinguish the normal ones from OOD
(or adversarial) ones. The underlying notion is that OOD (or adversarial) samples commonly deviate from the manifold
of normal ones, thus have high uncertainty. We compute and report the AUC of such a binary classification (L301).

**Q: Track the values of var($\mu$).** Thanks for the advice. We sampled 1000 training images from CIFAR-10 and
computed var($\mu$) of the last BN layer of a NSA and a NSA-i trained given $S = 5000$ architectures. We calculated the
average variance over all the channels and spatial locations, and the results of NSA and NSA-i are 0.00214 and 0.00082,
respectively, which testify the effectiveness of NSA-i. We'll track the full dynamics of var($\mu$) in the final version.

**Q: Ensemble gain of NSA-id.** At first, we clarify the ensemble gain of NSA-id is substantially more evident than that
of NSA-i. We have also discussed the potential reasons of the quick saturation of ensemble accuracy in L221-224. In
short, the introduced new parameters are rare, thus cannot adequately improve the weights diversity for ensembling.

**Q: Root cause of mode collapse.** Intuitively, the expectation w.r.t. architecture in NSA's training loss forces the shared
weights to be robust against architecture variability. Given such weights, the trained NSA may behave consistently
under diverse architectures, incurring mode collapse. To verify this, we assembled the 5 individuals with unshared
weights introduced in L277, and got striking 2.36% error and 0.003 ECE on CIFAR-10, confirming the above intuition.

**Q: Extension to DARTS search space.** Although the investigated search space is simpler than that in DARTS, the
issues of BN and weight sharing are shared between the spaces, and are observed frequently by the NAS community
[46, *2]. We think the discovered phenomena and proposed solutions are insightful for general NAS, while a systematic
investigation on general NAS is one of our future work.

**Q: Training time.** We clarify that we didn't perform searching. NSA's training time is almost identical to that of
WRN-28-10[†], e.g., 0.6 day on a GTX 2080Ti for 300 epochs (L115-116). The additional computations induced by the
complicated connections are only summations in the aggregation modules, which are negligible as compared to the
time-consuming convolutions.

**Q: Comparison to DARTS and ENAS.** DARTS and ENAS build networks with the parameter-efficient separable
convolutions, while NSA adopts the regular convolutions following WRN. Thus, comparing NSA with DARTS and
ENAS in the aspect of parameter number is not fair. Currently, the comparable baselines WRN-28-10[†] and *Average of*
*individuals* are outperformed by NSA evidently. And we leave the application to DARTS space as future work.

**To R2: Q: Extension and broader impact of NSA.** Thanks for the advice. We'll try to extend NSA to regression
tasks for uncertainty quantification and improve the broader impact in the final version. We'll add the NeurIPS paper.

**To R3: Q: Regarding $p(\alpha)$.** As stated in L98-101, $p(\alpha)$ is a uniform distribution over $S$ randomly pre-fetched
architectures by the ER-0.3 model. $p(\alpha)$ affects the architecture samples in the training (Eq. 3). When $p(\alpha)$ has larger
support, the optimized weights may be more helpful for architecture generalizing, but more under-fitting (see Table 1).

**Q: Regarding Eq.1 and Eq.3.** Eq.1 is the loss commonly used for training network with stochastic architectures, as in
SNAS [44], and the sampled architecture $\alpha$ is shared among all the instances in the mini-batch. Eq.3 uses instance
specific architectures to compute the training loss, namely, sampling an individual architecture $\alpha_i$ for each instance
$(x_i, y_i)$ in the mini-batch. The loss in Eq.3 is averaged over all the instances.

**Q: Regarding Figure 4.** The performance drop in Figure 4 may stem from the facts that the 500 used architectures are
randomly sampled and we perform only uniform ensemble instead of weighted ensemble. Thus assembling more base
learners may not give rise to rigidly better predictions. But the overall trend of NSA-id is substantially superior to that
of NSA-i. At last, we clarify that the first five ensemble accuracies of NSA-id in Figure 4 are 0.9613, 0.9648, 0.9658,
0.9659, 0.9659, while those of NSA-i are 0.9616, 0.9641, 0.9635, 0.9634, 0.9636. The comparisons confirm the claim
"ensemble gain is more obvious compared to NSA-i".

[*1] Takeru Miyato and Masanori Koyama. cGANs with Projection Discriminator.

[*2] Zhang et al. Deeper Insights into Weight Sharing in Neural Architecture Search.


[Meta-Review · NeurIPS 2020]

The authors propose an approach to addressing mode collapse and train/test disparity in deep neural networks, the work includes a good empirical evaluation. The reviewers make a number of suggestions of how the text could be improved which I encourage the authors to take on board.